

# Correction factors for prey size estimation from PenguCams

Owen Dabkowski[1], Ursula Ellenberg[1,2,3], Thomas Mattern[2,3,4], Klemens Pütz[5] and Pablo Garcia Borboroglu[2,3,6]

[1] Department of Marine Science, University of Otago, Dunedin, New Zealand
[2] The Tawaki Trust, Dunedin, New Zealand
[3] Global Penguin Society, Chubut, Argentina
[4] Department of Zoology, University of Otago, Dunedin, New Zealand
[5] Antarctic Research Trust, c/o Zürich, Zürich, Switzerland
[6] Centro Nacional Patagónico (CONICET), Chubut, Argentina

Corresponding author
Owen Dabkowski,
ob.dabkowski@gmail.com

## ABSTRACT

The use of animal-borne cameras enables scientists to observe behaviours and interactions that have until now, gone unseen or rarely documented. Researchers can now analyse prey preferences and predator-prey interactions with a new level of detail. New technology allows researchers to analyse prey features before they are captured, adding a new dimension to existing prey analysis techniques, which have primarily relied on examining partially or fully digested prey through stomach flushing. To determine prey size, the video footage captured needs a correction factor (pixel:mm ratio) that allows researchers to measure prey dimensions using image measuring software and convert the pixels to actual measurements. This in turn will help estimating the prey energy content. This method requires a reference object with known dimensions (such as beak measurements) to ground truth your distance. Using PenguCams we determined the correction factor by measuring a 2 cm section of 1 mm grid paper from video footage taken at known distances (10, 20, 30, 40, 50, 60 cm) in different salinities ranging from air and fresh water, up to 35 psu in 5 psu increments while controlling for temperature and pressure. We found no significant difference between correction factors of water at different salinities. However, due to their considerable differences in refraction index, correction factors contrast between water and air. Linear equations modelled from correction factors at tested distances help predict correction factors between tested distances and, therefore, enable a wider application of this research. We provide examples from PenguCam footage taken of Humboldt (*Spheniscus humboldti*), Tawaki (*Eudyptes pachyrhynchus*) and King (*Aptenodytes patagonicus*) penguins to illustrate the use of identified correction factors. This study provides a tool for researchers to further enhance their understanding of predator-prey interactions.

## INTRODUCTION

Advancements in technology such as GPS, temperature and salinity loggers have made it possible to track marine animals during their daily activities accurately (*Kooyman, 2004*; *Rutz & Hays, 2009*; *Wilmers et al., 2015*), allowing detailed analysis of behavioural patterns,

habitat utilization and feeding events (*Naito, 2004*; *Block, 2005*; *Chmura, Glass & Willaims, 2018*). However, questions still remained around specific details of feeding behaviour such as prey selection and foraging strategies (*Machovsky-Capuska et al., 2016*). With the development of animal-borne cameras scientists are now able to document the daily activities of animals, observe their behaviour when faced with different situations (*Mattern et al., 2018*; *Mattern & Ellenberg, 2022*) and determine specific predator–prey dynamics (*Machovsky-Capuska et al., 2016*). Moreover, this new technology enables researchers to examine prey features before capture, introducing a new dimension to current prey analysis methods, which have largely depended on studying partially or fully digested prey. Recent studies have used animal borne video cameras to conduct underwater surveys using White sharks (*Carcharodon carcharias*) and Grey Reef sharks (*Carcharhinus amblyrhynchos*) (see *Chapple et al., 2021*), investigate how prey size selection of Chinstrap penguins (*Pygoscelis antarcticus*) differs with various water conditions (see *Kuhn et al., 2022*), observe predator prey interactions between Tiger sharks (*Galeocerdo cuvier*) and sea turtles (see *Ryan et al., 2022*), and quantify diet of Northern Elephant seals (*Mirounga angustirostris*) (see *Yoshino et al., 2020*). As camera technology has gotten smaller and lighter, the range of animal species that can be studied using cameras has increased (*Moll et al., 2007*; *Naganuma et al., 2021*; *Kuhn et al., 2022*; *Pelletier et al., 2023*; *Mullaney et al., 2024*).

This is due to studies such as *Mattern et al. (2018)* developing their own animal-borne camera (PenguCams) to fit their target species, the yellow-eyed penguin (*Megadyptes antipodes*) (https://vimeo.com/showcase/4103142). These cameras have allowed to view prey capture events and get a better understanding of predator–prey dynamics and prey preference (*Mattern et al., 2018*; *Mattern & Ellenberg, 2022*). Since the creation of the penguin specific cameras, they have been deployed on a number of different species including Humboldt penguins (*Spheniscus humboldti*) (*Ellenberg et al., 2023*), Fiordland crested penguins (*Eudyptes pachyrhynchus*) (*Mattern & Ellenberg, 2021*; *Hornblow, 2022*), King penguins (*Aptenodytes patagonicus*; *Pütz & Cherel, 2023*), Erect-crested penguins (*Eudyptes sclateri; Mattern, 2023*), Magellanic penguins (*Spheniscus magellanicus*) and Gentoos penguins (*Pygoscelis papua*) (*Harris et al., 2023*), and African penguins (*Sphenuscus demersus*; *Glencross et al., 2024*).

While the cameras have been used to look at prey capture events (*Mattern et al., 2018*), one so far unused feature is the ability to estimate prey size to determine prey energy content. *Kalam & Urfi (2008)* have developed a method of prey size estimation for painted storks (*Mycteria leucocephala*) where they were positioned with a camera at a known distance and used the known beak size of birds to estimate prey size. The key part to this technique is the use of a reference object with known dimensions which allow you to ground truth your measurements. Alternatively, *Cavallo et al. (2020)* used fecal matter to work out overall biomass contribution to diet estimates. Using video footage, we now have a reliable method to determine size of prey and frequency of capture events. This provides new insights into predator–prey interactions and potential reasons for specific predator behaviour.

The aim of this project was to create a correction factor for animal-borne cameras.

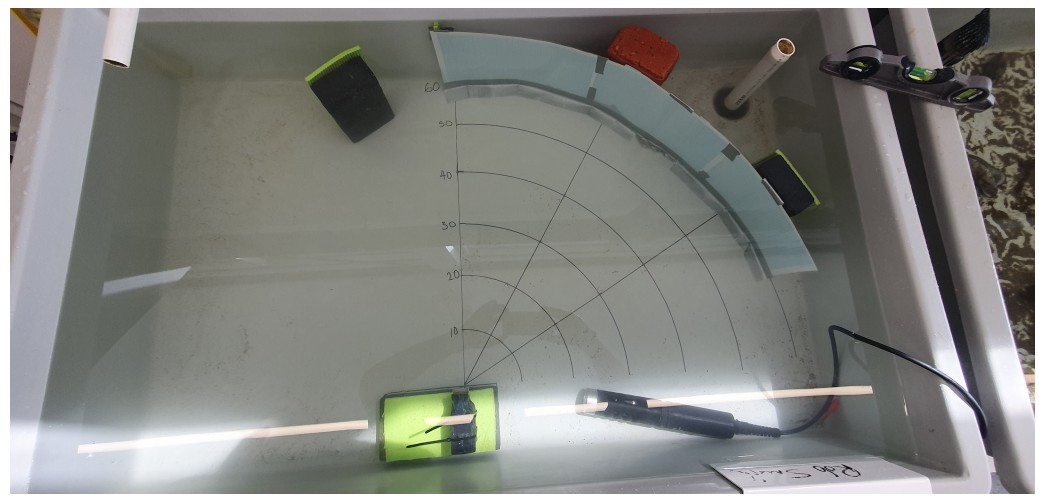

**Figure 1** **Experimental tank setup showing all instruments used.** Tank setup showing the stacked cameras on a brick (bottom left), HI98194 multiparameter (bottom right), quarter-circles drawn at different distances (10, 20, 30, 40, 50, 60 cm), lines at 0°, 33.75° and 67.5° and one mm grid paper placed at 60 cm.

## MATERIALS & METHODS

### Tank setup

The experiment was conducted in a $70 \times 100 \times 30$ cm tank. A brick was placed on the front edge where the cameras would be stacked. Then a grid was drawn on the tank with lines 10, 20, 30, 40, 50 and 60 cm away from the brick in a semi-circle pattern (Fig. 1) to represent the known range of a penguin's beak. Lines were drawn over the whole distance at an angle of 0°, 33.75° and 67.5° to help determine the field of view (FOV).

We used PenguCam (Eudyptes Ltd., Dunedin, NZ, https://pengu.cam/) cameras (lens: wide angle, 135°; video resolution: 1080p/30 fps, dimensions (LxWxH) $85 \times 36 \times 23$ mm, mass: 90 g), which were stacked on top of each other and secured so the lenses sat in line. These were placed so the center of the lens sat on the 0° line (Fig. 1). However, these methods can be applied to other devices.

### Experimental procedure

While cameras were recording, laminated one mm grid paper was placed along the 10 cm curve in front of the cameras and secured using duct tape and bricks after being straightened using a level. This was left for a few seconds to allow good video footage before it was moved backward to the 20 cm curve, and the process of straightening was repeated. This method was repeated at all distances in air before the tank was filled with fresh water and subsequently water of increasing salinities (5 psu up to a max of 35 psu, with separate camera recordings being taken for each salinity level). Increased salinity was achieved by mixing filtered seawater (~32 psu) with freshwater until the desired psu was reached. To reach 35 psu an appropriate amount of Aquavitro salinity for reefs was added. Temperature and psu were monitored using a HI98194 multiparameter and experiments

were all completed on the same day to control temperature and other potential variables out of our control. Covering a range of values will allow the results to be applicable to all penguin species that fit within the size range from Little Blue penguins (*Eudyptula minor*) to King penguins.

### Analysis

BORIS software (*Friard & Gamba, 2016*) was used to analyse video footage at the different distances from the camera. Using the inbuilt measurement tool, a two cm section of the grid paper was measured at three angles, 0°–Center, 20°–Halfway, and 40°–Edge in water and 0°–Center, 37.5–Halfway, and 65°–Edge in air, to work out pixel: mm ratio (hereafter referred to as correction factor). This process was repeated at all distances for all salinities and the ratios were recorded. If the photos taken were too blurry or bright, they were converted to high definition black and white using Microsoft Photos to make lines stand out and accurately measure a two cm distance. The mean correction factor of the three cameras was then calculated for every angle, distance, and salinity.

The data produced was analysed in R (version 4.3.2 in RStudio; *R Core Team, 2024*) using the 'car' (*Fox & Weisberg, 2019*) and 'dplyr' packages (*Wickham et al., 2023*). Mean correction factor was log transformed due to being right skewed and having unequal variance. To test the effect of salinity and angle a two-way ANOVA with distance as a blocking variable was run (mean correction factor ~Angle*psu + Distance). Following this, a post-hoc TukeyHSD test to show whether salinity or angle has a significant effect on the correction factor.

The mean correction factor for each salinity measurement was calculated ($n = 3$) and plotted with error bars to demonstrate differences (Fig. 2). As salinity was not significant, the data was grouped into water and air when graphs were created (Fig. 3). Water and air graphs plotted mean natural-log correction factors against natural-log distance with linear models of the relationships created for each angle (Table 1) ($n = 3$). The linear models were fitted with 95% confidence intervals (CI). This provides equations to calculate correction factors in between measured distances. All figures were made using ggplot2 (*Wickham, 2016*) and Fig. 3 was created using 'gridExtra' (*Auguie, 2017*), 'grid' (*R Core Team, 2024*), and 'gridtext' (*Wilke & Wiernik, 2022*) packages.

## RESULTS

Salinity (two-way ANOVA: $F_8 = 38.45$, $p < 0.0001$) and angle (two-way ANOVA: $F_2 = 24.603$, $p < 0.0001$) had a significant effect on correction factor. However, the Tukeys HSD ($p = 0.05$) revealed that significant results were driven by the difference in correction factor between air treatments and water treatments, meaning there was no difference in correction factor between water treatments with different salinities (Fig. 2). Because of this, the dataset and graphs were simplified to water *vs* air (Fig. 3). Furthermore, the Tukey's test revealed that there was no significant difference between the center and halfway measurements, so these were combined resulting in two angle measurements (Center and Edge). The interaction between salinity and angle was non-significant (two-way ANOVA: $F_{16} = 3.87$, $p = 0.0586$).

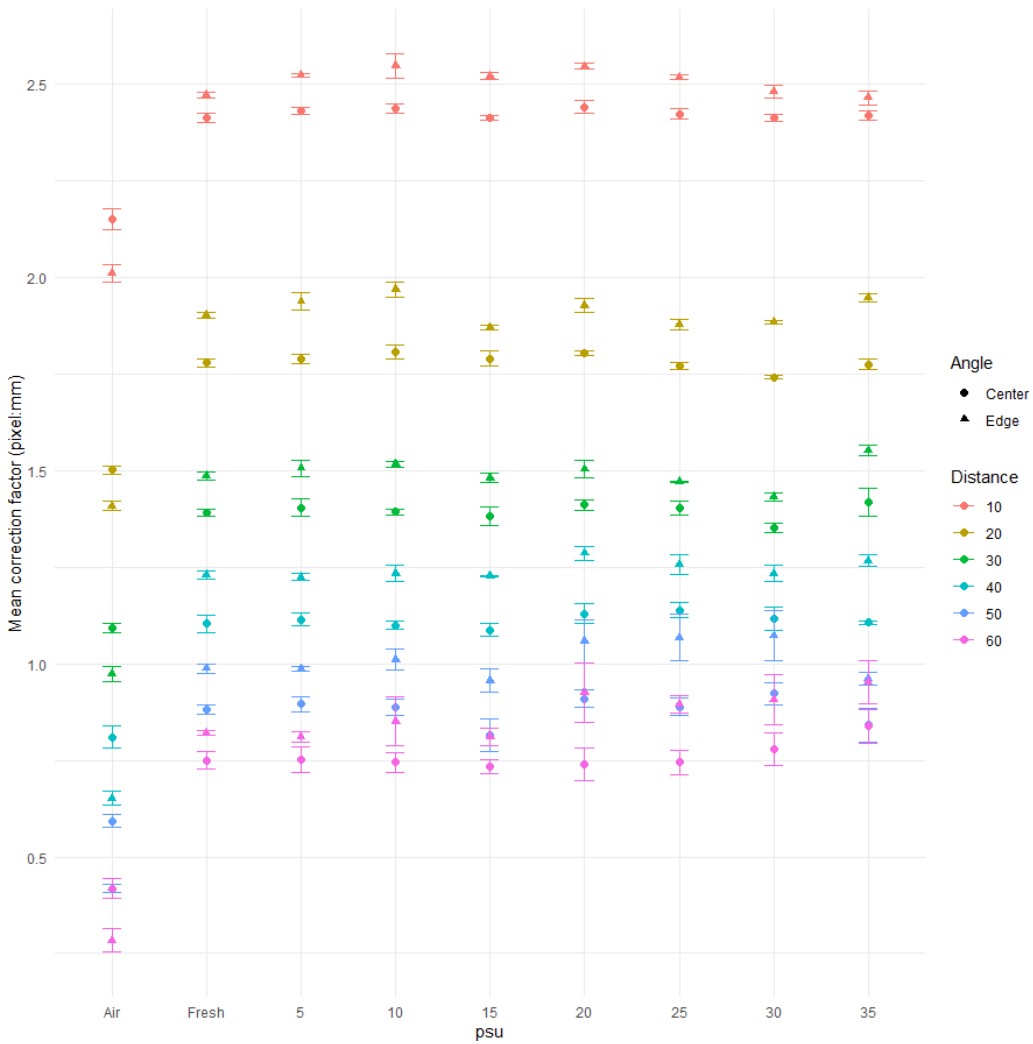

**Figure 2  Scatterplot showing correction factor at each distance and each salinity.** Scatter plot showing the correction factor (pixel: mm) at each salinity level. Colours represent different distances. Shape represents the angle (center or edge) where the measurement was taken. Error bars show standard error.

Figure 3 shows that the correction factor followed a negative linear trend with center having a larger correction factor than edge in air, and edge having a larger correction factor than center in water. Overall, the correction factor was larger in water compared to air for both angles as was expected. Table 1 provides the equations for the linear models used in Fig. 3. Table 2 provides a quick non-natural log transformed correction factor guide to use for set distances, however the models in Table 1 can be used for predictions made in between those distances.

## DISCUSSION

For precise prey measurements from animal borne camera footage, a correction factor must be created for known distances. Here we used a range of plausible distances from the
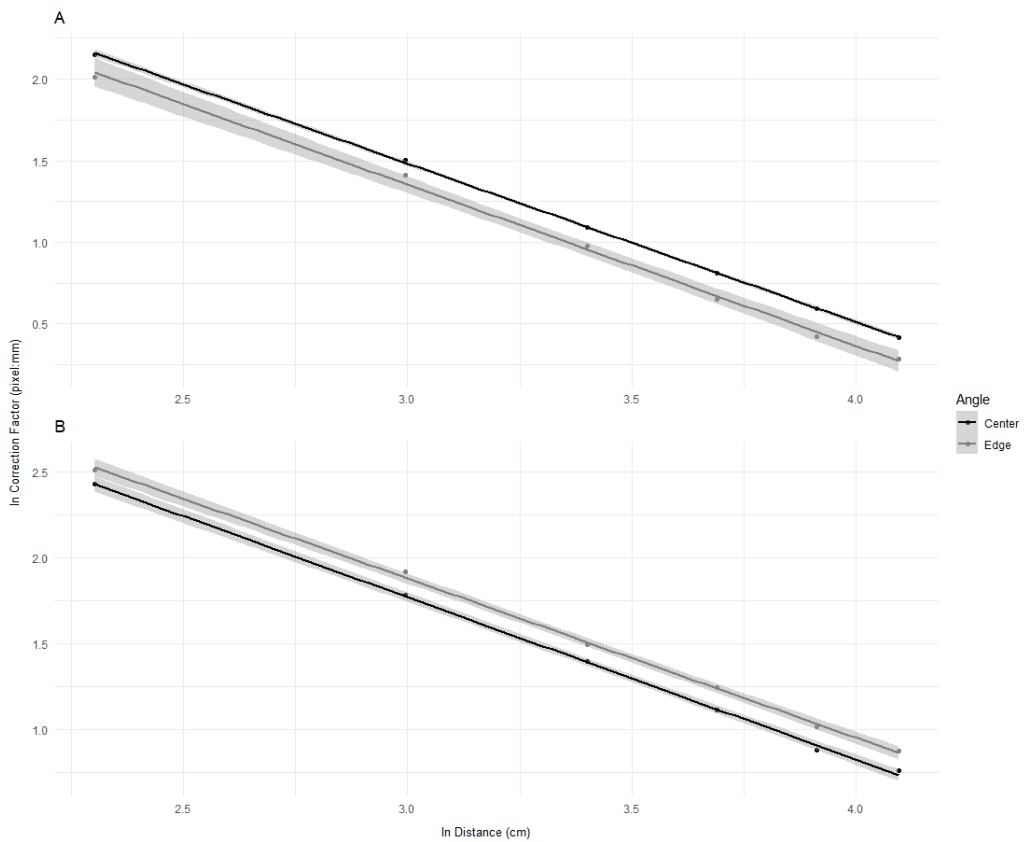

**Figure 3** **Linear equation showing the decrease in natural log correction factor over an increase in natural log distance from camera for the edge and center of photo.** Scatter plot showing the effect of distance (cm) on correction factor (pixel: mm) for two angles (Center and Edge) in air (A) and water (B). Each angle is fitted with a linear model and shaded with a 95% confidence interval. Both distance and correction factor have been transformed using natural log transformations (ln).

**Table 1** **Linear equation to calculate correction factors with $R^2$ values for different locations within the image and different mediums.** List of equations for the linear models plotted in Fig. 3 with the $R^2$ values.

|  | $R^2$ value | Equation |
|---|---|---|
| Air center | 0.999 | $\ln_{\text{Correction Factor}} = 4.369631 - 0.97117\,(\ln_{\text{Distance}})$ |
| Air edge | 0.997 | $\ln_{\text{Correction Factor}} = 4.31735 - 0.98806\,(\ln_{\text{Distance}})$ |
| Water center | 0.999 | $\ln_{\text{Correction Factor}} = 4.59898 - 0.94303\,(\ln_{\text{Distance}})$ |
| Water edge | 0.999 | $\ln_{\text{Correction Factor}} = 4.65661 - 0.92583\,(\ln_{\text{Distance}})$ |

camera lens to the penguin's beak. Plausible distance was informed by coauthor experience and measurements taken in the field. We found no differences in correction factors at different salinities. Therefore, we provide two sets of correction factors, one for air, another for water. *Quan & Fry (1995)* and *Austin & Halikas (1976)* found a significant effect of salinity on refraction rate, however, they were measuring down to 0.001 changes. *Austin & Halikas (1976)* found that over a range of salinity from 0–40 psu the index of refraction

**Table 2 Correction factor for set distances in different mediums at different locations within photo.** Correction factor (pixel:mm) at different distances (10, 20, 30, 40, 50, 60) for air and water in the center and edge of photo (in bold) ±95% CI.

|  |  | 10 | 20 | 30 | 40 | 50 | 60 |
|---|---|---|---|---|---|---|---|
| Air | Center | **8.60** ± 0.399 | **4.49** ± 0.079 | **2.99** ± 0.061 | **2.25** ± 0.110 | **1.81** ± 0.051 | **1.52** ± 0.066 |
|  | Edge | 7.47 ± 0.285 | 4.09 ± 0.086 | 2.65 ± 0.088 | 1.92 ± 0.056 | 1.52 ± 0.029 | 1.33 ± 0.071 |
| Water | Center | **11.30** ± 0.242 | **5.95** ± 0.166 | **4.04** ± 0.153 | **3.04** ± 0.109 | **2.42** ± 0.133 | **2.14** ± 0.138 |
|  | Edge | 12.32 ± 0.465 | 6.79 ± 0.275 | 4.11 ± 0.174 | 3.48 ± 0.114 | 2.76 ± 0.212 | 2.40 ± 0.217 |

increased by 0.01 from 1.335 to 1.345. In our experiment, a change this small would be lost in the overall error generated from taking measurements by hand using video footage, therefore, difference in refraction between salinities are insignificant for our purposes. As a result, we only found a difference in the correction factor between water and air which is explained by water having more particles to refract light compared to air (*Quan & Fry, 1995*). The correction factor is larger on the edge than in the center in water which is known as barrel distortion (*Harry Ng & Kwong, 2009*). In air the opposite trend is observed which is known as pincushion distortion (*Ayinde et al., 2011*). Differing distortion occurs because the refractive index of water is higher than air (*Andrews, 2023*). Meaning light traveling into the camera will refract differently when the camera is underwater and therefore has a water-air interface *versus* in air where there is no interface, resulting in different angular magnifications and therefore different distortions (*Andrews, 2023*).

We provide a reference table for correction factors at a given distance to quickly measure and estimate size of an item when analysing PenguCam footage. Figure 4 demonstrates how the correction factor can be used to determine the length of captured fish using an example from Tawaki footage taken using the PenguCams. Distance between the camera lens and tip of the beak was measured in the field when cameras were attached which allows the correction factor for the known distance to be used. The fish in the image is estimated to be 7.27 mm long provided the distance from the camera lens to the beak is ~30 cm.

For distances outside of the reference table (Table 2), the linear models fitted to the data can provide the desired correction factor (Table 1). As they are natural log transformed the outputs need to be exponentially transformed, *i.e.*,

$$\text{Correction factor} = e^{(\text{ln transformed correction factor})}$$

in order to be used. Figure 5 illustrates how the known size of the bird's beak can be used to determine the distance between it and the camera lens when the distance is unknown such as when the bird lunges for prey or turns their head. In this case, the beak depth is 21.5 mm which when measured in BORIS was 116.6 pixels. This corresponds to a correction factor of 5.42 pixels: mm. Comparing this with reference table we know that the distance must be between 10 and 20 cm and we can use the linear model for Air Center to predict that the distance is 15.8 cm.

Measurements can also be taken of other species that the penguins interact with to work out prey overlap or competitive ability. Figure 6 demonstrates interaction with and estimated size of a slender tuna (*Aptenodytes patagonicus*) that is competing with the King penguin for food. In this example the tuna is 972.1 pixels long and 230.9 pixels tall at a

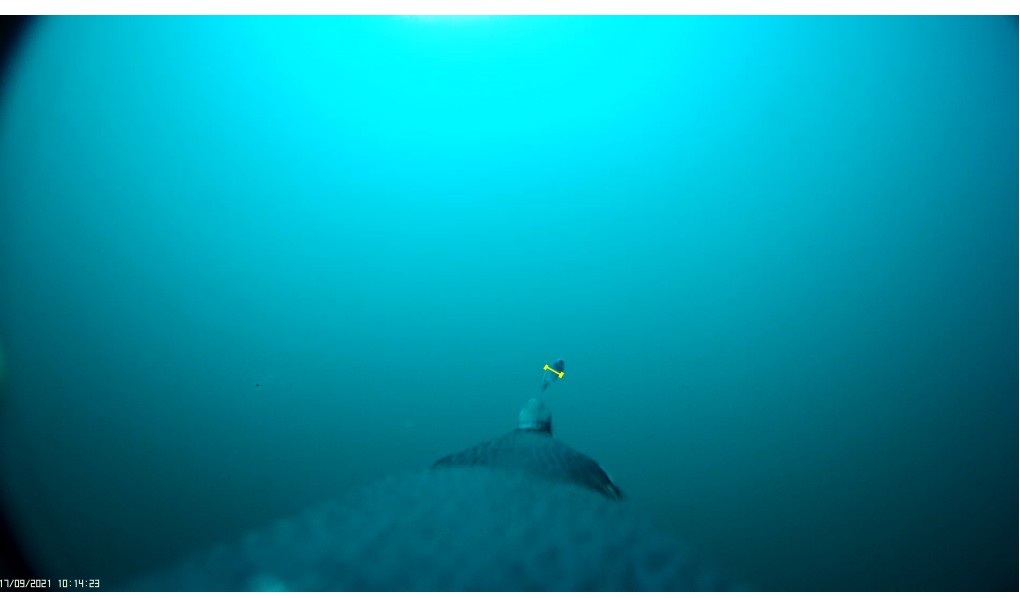

**Figure 4** **Image of larvae capture from Tawaki PenguCam footage with green line showing area measured to determine size.** Image of a fish larvae captured by a Tawaki (*Eudyptes pachyrhynchus*) in Milford Sound. Line measuring height in pixels.

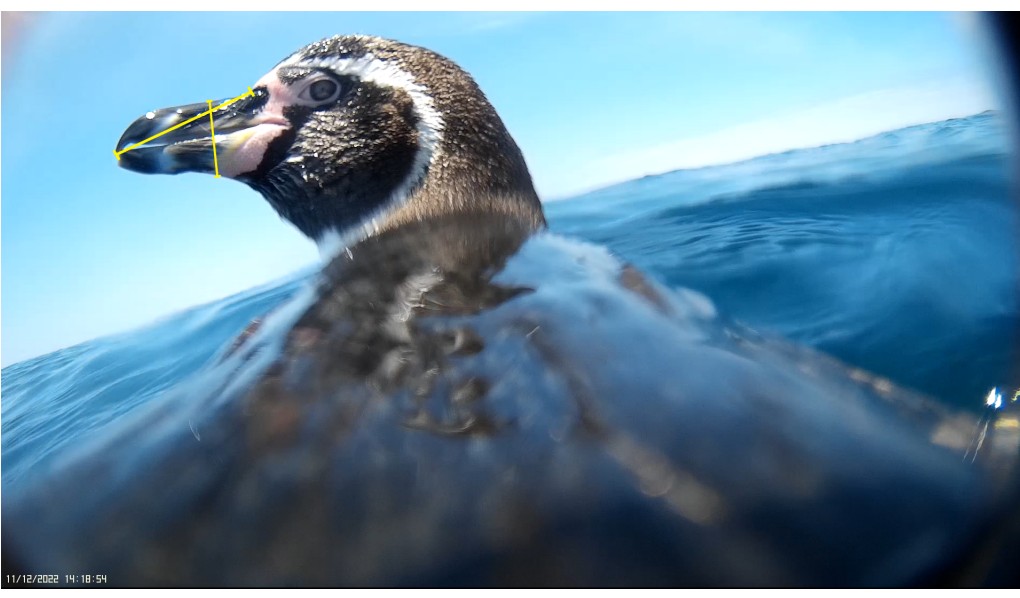

**Figure 5** **Side profile image of Humboldt penguin allowing for beak measurements to be taken.** Image of a female Humboldt penguin (*Spheniscus humboldti*) off Isla Choros, Humboldt Archipelago, Chile. Lines measuring beak depth (top of beak to bottom of beak) and length (back to tip of beak) in pixels.

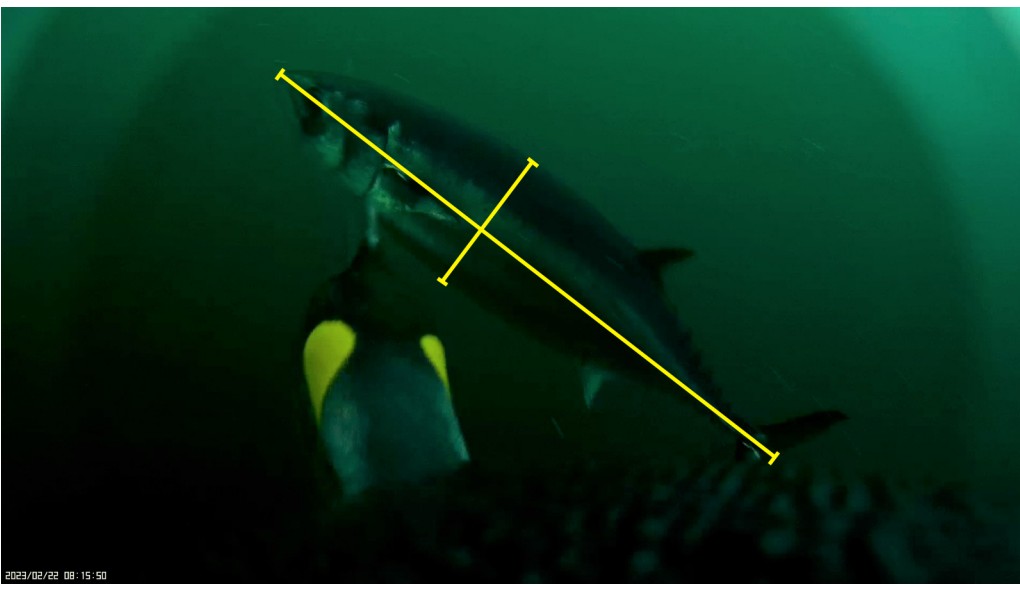

**Figure 6** **Image of King penguin pecking at a slender tuna during a feeding frenzy.** Image of a female King penguin (*Aptenodytes patagonicus*) pecking at a tuna (*Allothunnus fallai*) during a bait ball feeding frenzy off Tierra del Fuego, Strait of Magellan. Lines measuring total length (long line) and height (short line) in pixels.

distance of 40 cm. Therefore, a correction factor of 3.04 is used to estimate a length of 319 mm long and 76 mm tall. However, the distance from the camera can easily change when penguins lunge for prey. If beak length and depth is known, then it can be used to determine the exact distance, and the linear model can be used to calculate the correction factor if it sits outside of the reference table values. Without a reference measurement or a known distance, it will not be possible to measure the size of prey as the correction factor cannot be determined.

In the field we recommend taking measurements of bill length and depth which will allow for ground truthing of the correction factor or calculating distance from camera if the penguin lunges or turns its head. Furthermore, knowing the distance from the camera to the tip of the beak will help to speed up the measurement process. Using these correction factors, researchers can estimate the size of competitors and assess the energy content of prey to determine the specific energy gain an animal receives from each prey item.

## CONCLUSIONS

This study provides a crucial advancement in the field of marine biology by enabling precise prey size estimations from animal-borne video footage, significantly enhancing our ability to study predator–prey interactions and energy dynamics in marine ecosystems. Animal-borne video cameras offer unique insights that reveal added information about an animal's behaviour. In this study we expanded on their uses by creating a correction factor which allows for size estimates to be taken from video footage. Our work along with effort required to obtain said prey item (see *Mattern et al., 2018*;

*Petrovski, Sutton & Arnould, 2023*) will explain why animals target certain prey items over others, or how much energy they gain in a single feeding period compared to how much energy they expended. We trust this will help unlock the full potential of animal borne video camera footage, thus, enabling a new, unique perspective when studying not only diet composition but also foraging behaviour that reveals the previously unseen interactions which take place underwater.

## ACKNOWLEDGEMENTS

We thank Adam Brook and Linda Groenewegen at Portobello Marine Laboratory for providing access to facilities and equipment. We are grateful to the Tawaki Trust for providing the PenguCams for the experiments and Sally Carson for providing reef salt. Thank you to Lars Ellenberg for his sharp mind and great feedback during the early stages of this project and Forest Chaput de Saintonge for his knowledge of camera distortion.

### Funding

This work was supported by the Marine Science Department Summer Research Bursary (UO-00701437) provided by Miles Lamare to Owen Dabkowski. The funders had no role in study design, data collection and analysis, decision to publish, or preparation of the manuscript.

### Grant Disclosures

The following grant information was disclosed by the authors:
Marine Science Department Summer Research Bursary: UO-00701437.

### Competing Interests

Klemens Pütz is the scientific director, co-founder and trustee of the Antarctic Research Trust. Pablo Garcia Borboroglu is the Founder and President of the Global Penguin Society and Patron of the Tawaki Trust. Ursula Ellenberg and Thomas Mattern are founding members and trustees of the Tawaki Trust, and representatives for Global Penguin Society.

### Author Contributions

- Owen Dabkowski conceived and designed the experiments, performed the experiments, analyzed the data, prepared figures and/or tables, authored or reviewed drafts of the article, and approved the final draft.
- Ursula Ellenberg conceived and designed the experiments, performed the experiments, authored or reviewed drafts of the article, and approved the final draft.
- Thomas Mattern conceived and designed the experiments, authored or reviewed drafts of the article, and approved the final draft.
- Klemens Pütz conceived and designed the experiments, authored or reviewed drafts of the article, and approved the final draft.
- Pablo Garcia Borboroglu conceived and designed the experiments, authored or reviewed drafts of the article, and approved the final draft.

## Data Availability

The raw data is available in the Supplemental File.

## Supplemental Information

Supplemental information for this article can be found online at http://dx.doi.org/10.7717/peerj.18598#supplemental-information.

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
