# Peer review of "Correction factors for prey size estimation from PenguCams"

_PeerJ, doi:10.7717/peerj.18598_

## Round 0.1 · original submission · Minor Revisions

Both reviewers agree the paper is well written, but both provide suggestions for improvement. Reviewer 2's comments on statistical presentation should be carefully addressed in the revised manuscript.

Reviewer 1 ·

Basic reporting

OK

Experimental design

OK

Validity of the findings

OK

Additional comments

This is well-designed research project with a clear outcome for future use. My concerns are all minor. For example, in the abstract and introduction, you should include the need for a reference object of known dimension as a critical piece of information to have in order to estimate prey sizes. In this case, you’ve suggested that the beak can be used. Adding that key detail is important. Also important to stress is that the measurement of prey size is the video is limited to those prey that are at the same distance from the camera as the beak (or other known reference object captured in the image). Because you can’t directly measure the distance to free swimming prey in video, the reference length only applies at the known distance. Acknowledging that constraint will be helpful, and further emphasizes your suggestion to collect beak and other morphometric measurements that could assist in the estimation of prey dimensions of ingested prey.
A second consideration is that, while you focus exclusively on PenguCams, there are other recording options available, and other studies using other cameras that could be referenced as examples of studies that examine predator-prey interactions, quantify prey capture events, or attempt to characterize prey, their aggregations, density, etc. Adding a few other examples would simply expand the scope of work considered and highlight for readers the broader research effort currently underway on this important topic.

Reviewer 2 ·

Basic reporting

No comment.

Experimental design

The methods are not clearly described and lack information to review if analyses are sound. See my comments below.

Validity of the findings

See comment below concerning statistical analysis. Code is not provided (only the raw data), so that I cannot check if the results are reproducible.

Additional comments

Additional comments can be found below.

Annotated reviews are not available for download in order to protect the identity of reviewers who chose to remain anonymous.

---

## Round 0.2 · accepted · Accept

Thank you for addressing the reviewers' comments. The MS is ready for publication.